# Evaluation of Intradermal PRRSV MLV Vaccination of Suckling Piglets on Health and Performance Parameters under Field Conditions

**DOI:** 10.3390/ani13010061

**Published:** 2022-12-23

**Authors:** Georgios Maragkakis, Labrini V. Athanasiou, Serafeim C. Chaintoutis, Dimitra Psalla, Polychronis Kostoulas, Eleftherios Meletis, Georgios Papakonstantinou, Dominiek Maes, Georgios Christodoulopoulos, Vasileios G. Papatsiros

**Affiliations:** 1Clinic of Medicine, Faculty of Veterinary Medicine, School of Health Sciences, University of Thessaly, 43100 Karditsa, Greece; 2Diagnostic Laboratory, School of Veterinary Medicine, Faculty of Health Sciences, Aristotle University of Thessaloniki, 54627 Thessaloniki, Greece; 3Laboratory of Pathology, School of Veterinary Medicine, Faculty of Health Sciences, Aristotle University of Thessaloniki, 54124 Thessaloniki, Greece; 4Laboratory of Epidemiology & Artificial Intelligence, Faculty of Public Health, School of Health Sciences, University of Thessaly, 43100 Karditsa, Greece; 5Department of Internal Medicine, Reproduction and Population Medicine, Faculty of Veterinary Medicine, Ghent University, B-9820 Merelbeke, Belgium; 6Department of of Animal Science, Agricultural University of Athens, 75 Iera Odos Street, Votanikos, 11855 Athens, Greece

**Keywords:** intradermal, PRRSV, MLV, vaccine, BW, ADG, pig

## Abstract

**Simple Summary:**

The aim of this study was to investigate the effect of an intradermal (ID) administration of a PRRSV-1 modified live virus (MLV) vaccine in comparison to an intramuscular (IM) administration on health and performance. A total of 187 suckling piglets of a PRRSV-positive commercial farrow-to-finish farm were assigned to four groups: group A—PRRSV ID, group B—PRRSV IM, group C—control ID, and group D—control IM. At 2 weeks of age, all of the study piglets were either vaccinated with a PRRSV-1 MLV vaccine or injected with the vaccine adjuvant (controls). The collected blood serum samples were tested by ELISA and qRT-PCR. The side effects, body weight (BW), average daily gain (ADG), mortality rate, and lung and pleurisy lesions scores (LLS, PLS) were also recorded. Our study demonstrated that the ID vaccination of suckling piglets with a PRRSV-1 MLV vaccine has a positive effect on the piglets’ health and performance, including an improved BW and a lower LLS and PLS index at their slaughter, as well as a decreased mortality rate at the growing/finishing stage.

**Abstract:**

Porcine reproductive and respiratory syndrome virus (PRRSV) causes respiratory disease in weaning and growing pigs. A vaccination against PRRSV is one of the most important control measures. This trial aimed to evaluate the effect of the intradermal (ID) administration of a PRRSV-1 modified live virus (MLV) vaccine in comparison to the intramuscular (IM) administration on the piglets’ health and performance. A total of 187 suckling piglets of a PRRSV-positive commercial farrow-to-finish farm were assigned to four groups: group A—PRRSV ID, group B—PRRSV IM, group C—control ID, and group D—control IM. At 2 weeks of age, all the study piglets were either vaccinated with a PRRSV-1 MLV vaccine or injected with the vaccine adjuvant (controls). The collected blood serum samples were tested by ELISA and qRT-PCR. The side effects, body weight (BW), average daily gain (ADG), mortality rate, and lung and pleurisy lesions scores (LLS, PLS) were also recorded. The ELISA results indicated that the vaccination induced an important seroconversion at 4 and 7 weeks. Significant differences in the qRT-PCR results were noticed only at 10 weeks in group A vs. group C (*p* < 0.01) and group B vs. group C (*p* < 0.05). High viral loads, as evidenced by the qRT-PCR Ct values, were noticed in animals of both non-vaccinated groups at 7, 10, and 13 weeks. An ID vaccination has a positive impact on the BW at the piglets’ slaughter, while both an ID and IM vaccination had a positive impact on the ADG. The mortality rate was lower in vaccinated groups at the finishing stage. The LLS and PLS were significantly lower in the vaccinated groups. In conclusion, our study demonstrated that the ID vaccination of suckling piglets with a PRRSV-1 MLV vaccine has a positive effect on the piglets’ health and performance, including an improved BW and a lower LLS and PLS index at their slaughter, as well as a decreased mortality rate at the growing/finishing stage.

## 1. Introduction

Porcine reproductive and respiratory syndrome (PRRS) induces mainly respiratory disease in weaning and growing pigs, as well as reproductive failures in the breeding stock [1,2,3]. Studies have shown the important financial impact which outbreaks of PRRS have on the global swine industry [1,4,5,6]. PRRS virus (PRRSV) is a key pathogen involved in the porcine respiratory disease complex (PRDC) [7,8], causing also major economic losses worldwide. PRRSV can achieve synergism with other respiratory bacteria, such as *Mycoplasma hyopneumoniae* (*M. hyo*) and *Actinobacillus pleuropneumoniae* (App), even in PRRSV-vaccinated farms [7,9,10].

The control of prevention strategies for a PRRSV infection includes mainly the vaccinations of sows and/or piglets with modified live virus (MLV) vaccines. In sows, killed vaccines (KV) are also used [1,11,12,13,14,15]. The control of PRRS by vaccination is important also to decrease economic losses due to a decreased performance and feed efficiency [7,16]. Studies on the effectiveness of the PRRSV vaccination at the suckling age under field conditions, including the investigation of its impact on the performance parameters of growing/finishing pigs, are rarely reported [17,18,19]. However, previous studies reported that a PRRSV vaccination is associated with an improvement in the growth performance, under field conditions [20,21].

Intradermal (ID) vaccination has gained increasing interest in swine medicine as an alternative to intramuscular (IM) vaccination due to the advantage of aiming at antigen-presenting cells in the epidermis close to the skin-draining lymph nodes [22,23]. Many safety and efficacy studies with an ID vaccination have been reported against various respiratory pathogens, such as PRRSV [24,25,26,27,28], porcine circovirus type 2 (PCV2) [29], Aujeszky’s disease [30], and *M. hyo* [31].

Previous studies showed that an ID vaccination against PRRSV has at least a similar immunological effect compared to IM [24,25,27]. Other studies also reported that an ID vaccination could provide a similar protective immunity compared with an IM vaccination, under experimental conditions [32]. Moreover, the ID administration of a commercial PRRSV MLV vaccine can induce efficient protection in pigs later exposed to a genetically diverse PRRSV strain, under field conditions [24]. Finally, an ID vaccination that is applied with intradermal needle-free devices has important benefits in terms of the animal’s welfare as a needle-free administration (e.g., a less invasive and less painful administration and the absence of histological lesions on the point of application), the prevention of the iatrogenic transmission of pathogens in vaccinated pigs, the elimination of accidental worker self-injections, and the carcass’ quality (e.g., a lack of needle-induced injection site lesions or broken needles and residual needle fragments in pork carcasses) [26,33,34,35,36,37]. Even though the ID vaccination of piglets against PRRSV is a common practice, there is no literature investigating, under field conditions, the effect of an ID vaccination of suckling piglets against PRRSV on the growth performance parameters up to the finishing stage.

This field study aimed to evaluate the impact of the ID vaccination of piglets with a commercial PRRSV MLV vaccine in comparison to an IM vaccination on their health status and performance until they reached their slaughter age, under field conditions, in a farm suffering from an active PRRSV infection.

## 2. Materials and Methods

### 2.1. Animal Ethics and Experimental Farm

All procedures regarding animal care, handling, treatment, and use were approved by the Ethical Committee of the Faculty of Veterinary Medicine, School of Health Sciences, University of Thessaly (permission number: 98/19-12-2019). The owner of the farm the research took place on was thoroughly informed and provided written consent for the use of the farm animals in the described experimentation.

The current trial was carried out in a farrow-to-finish commercial pig farm located in Central Greece. The productive capacity of the farm was 150 commercial hybrid sows (Large White x Landrace, Topigs Norsvin). A grandparent nucleus was maintained on the farm for producing breeding gilts. The herd practiced a 1-week batch production system. At weaning (28 ± 3 days), the grouping of the piglets at flat-deck nurseries was equally based on their body weight and sex, applying a stratified randomization procedure. The semen used for the artificial insemination was collected from commercial hybrid boars (Topigs Norsvin) that were maintained on the farm. 

The vaccination protocol of farm sows included routine vaccinations against PRRSV, Aujeszky’s disease virus, swine parvovirus, atrophic rhinitis, swine erysipelas, *Escherichia coli*, and *Clostridium perfringens*. Piglets were vaccinated against PCV2 and *M. hyo* at the age of 3 weeks. The routine antiparasitic program of sows included a single IM administration of ivermectin two weeks before farrowing. As for PRRSV, the sows were vaccinated following the 6th day of lactation and the 60th day of the gestation vaccination plan, the gilts were vaccinated twice with a three-week gap between the doses before entering productive life, and the boars were vaccinated biannually as a farm routine. For the immunization of the breeding stock, the Porcilis PRRS vaccine by MSD Animal Health was used, the same commercial product which was utilized in our trial. At the onset of the current experimentation, no piglets on this farm were ever vaccinated against PRRSV before.

The feed provided was homemade and based on corn/barley/wheat–soya, using an automatic feeding system. Animals participating in the trial had at-will access to drinking water and an automatic temperature and humidity system-controlled housing facility. 

### 2.2. Farm History-Pre-Trial Period

Four weeks prior to the onset of the experimentation, blood sampling was performed on different ages; sows and gilts (12 total blood samples from gilts and sows in gestation and lactation stages), suckling and weaned piglets, and growing and finishing pigs (8 blood samples at the ages of 2, 4, 7, 10, 13, 17, and 21 weeks). Testing via qRT-PCR revealed the farm’s positivity for PRRSV during the pre-trial period, with the following prevalence: 100% at 4, 7, 10, and 13 weeks, 58% at 17 weeks, and 22% at 21 weeks. Based on the complete ORF 5 sequencing and bioinformatics analysis, it was found that the PRRSV strain detected in the farm had a 90.7% sequence identity with the DV vaccine strain. The PRRSV genome was not detected at the age of 2 weeks. In addition, the examination of the blood samples from unvaccinated suckling piglets (7, 14, 21, and 28 days of age) for PRRSV-specific antibodies via ELISA showed low levels of maternal antibodies at 14 days of age and low levels or the absence of maternal antibodies at 21 and 28 days of age. 

The main clinical problems at the weaning and fattening stage were the increased morbidity associated with respiratory symptoms and an increased mortality rate, due mainly to secondary co-infections. Piglets belonging to batches born prior to the beginning of the field trial were vaccinated only against PCV2 and *M. hyo* at the age of 3 weeks. 

### 2.3. Experimental Animals

In total, 187 healthy suckling piglets (2 weeks of age) were included in the study and were assigned to 4 groups. Four replicates were formed in each group, in each of which 11–12 piglets were allocated), as shown in Table 1. The diseased piglets with a significantly low body weight (BW) or congenital flaws (e.g., hernia) were excluded from the trial. 

The piglets were placed individually (within the litters) in the test groups equally based on the BW and sex, as well as the parity number of their sows (parities 1 to 5). At the time of admission, the piglets were ear-tagged. Four unique numbered ear tags were used for their identification and the numbers were recorded. The color of the tags varied by the production batch and the lost tags were replaced. The pigs of the test and control groups were mixed in different pens in the same nursery room. Then, all of the pigs were housed as usual in shared pens in all production stages from nursery to finishing age.

### 2.4. Experimental Material, Dosage, and Administration

The vaccine utilized in the trial for the immunization of the piglets against PRRSV was Porcilis^®^ PRRS (MSD Animal Health). It is a commercially available, live attenuated lyophilized vaccine, containing per dose of 2 mL (IM application) or 0.2 mL (ID application) of reconstituted vaccine, the active substance (lyophilisate), i.e., the live attenuated PRRSV-1 strain DV at titers 10^4.0^–10^6.3^ TCID_50_ (50% tissue culture infective dose), along with the Diluvac Forte^®^ adjuvant (solvent) that contains dl-α tocopheryl acetate at a concentration of 75 mg/mL. 

The piglets included in the test groups (A and B) were vaccinated at the age of 2 weeks with 1 dose of Porcilis^®^ PRRS, diluted in 0.2 mL (group A, ID) or 2 mL (group B, IM) of Diluvac Forte^®^. The piglets included in the control groups (C and D) were not vaccinated against PRRSV and only Diluvac Forte^®^ was administered (0.2 mL for group C, or 2 mL for group D) (Table 1). An IDAL (IntraDermal Application of Liquids) vaccinating device was used for the needle-free intradermal injection of piglets belonging to ID groups (A and C). For the administration of the piglets belonging to the IM groups (B and D), an automatic syringe, with a fixed 2 mL volume, was used. A new, clean 18 G needle (size approx. 0.9 × 13 mm) was used for each IM trial group.

No other vaccinations or vaccines against PRRSV or other veterinary products (concurrently or simultaneously) were applied within 14 days before or 7 days after the administration of Porcilis^®^ PRRS. The piglets were vaccinated against PCV2 and *M. hyo* with a single dose vaccine at the age of 3 weeks. All the treatments of the growing–fattening pigs (preventive oral or injectable treatments and vaccinations) were recorded for the treatment and control groups.

### 2.5. Samplings and Laboratory Examinations

Blood samples were collected from the same 3 identified pigs belonging to each group for each of the 4 replicates. Sampling was performed at the ages of 4, 7, 10, 13, 17, and 21 weeks. Serological testing for PRRS was performed on each serum sample according to the manufacturer’s recommendations (IDEXX PRRS X3 Ab Test). Sample/Positive (S/P) ratios were used as the estimates of the antibody titers. Serum blood samples were also subjected to a nucleic acid extraction using the PureLink^®^ Viral RNA/DNA Mini Kit (Invitrogen). The extracts were tested for the PRRSV genome by qRT-PCR using a previously described TaqMan probe-based qRT-PCR method [38]. Ct (cycle threshold) values were used for the estimation of viremia titers. 

### 2.6. Records

#### 2.6.1. Local and Systemic Reactions

The investigators consistently checked all the study animals individually for reactions and side effects post-vaccination, on day 0 and at 1, 4, 7, and 14 days post-vaccination. That was a group observation that did not require the individual inspection and palpation of each piglet (which was not practically possible with a large number of animals in the study). 

#### 2.6.2. Performance Parameters 

To obtain the average daily gain (ADG) (g/pig/day), each different group per replicate was weighed on admission day, on weaning day, at the end of the nursery period, and at their slaughter age. The study animals were weighted in study groups and the calculation of the means as an individual measurement was not possible due to the farm routine. In addition, the age of slaughter was recorded.

#### 2.6.3. Mortality

At any time after an admission, a post-mortem examination was presented on a piglet that died or was culled. The percentage of mortality was recorded at different periods (nursing, growing, and finishing stages).

#### 2.6.4. Lung Lesion Score (LLS) and Pleurisy Lesion Score (PLS)

All the lungs of the pigs included in the trial were examined and scored for their typical gross lesions according to the method of Goodwin and Whittlestone (1993). The lesions of cranio-ventral pulmonary consolidation (LLS ranged from 0 to 55 at maximum) of each tested lung were the addition of the proportion of each lobe surface with signs of typical inflammation, multiplied by the weighting of each lobe. Moreover, the pleurisy lesions (PLS) were evaluated as follows: 0—no pleurisy lesions, 1—topical adhesions (spots), and 2—larger adhesions. To prevent bias, one pathologist responsible for the lung examination at the slaughterhouse was blinded for the treatment group of the animals.

### 2.7. Statistical Analysis and Study Power Analysis 

The data were presented as the mean ± standard deviation. A statistical analysis was done using IBM SPSS 21 (IBM Corp., Armonk, NY, USA). A data analysis among the groups was performed using the Kruskal–Wallis H test, and the Mann–Whitney U test was used for multiple comparisons. The comparisons within the groups were performed using the Friedman test and the Wilcoxon Signed Rank test. Significant differences were considered below the 0.05 level. A power analysis was done with G*Power software (version 3.1) [39]. The total sample size and power analysis were estimated for each one of the measured study variables (the ELISA, qRT-PCR results, body weight, etc.).

## 3. Results

### 3.1. Local and Systemic Reactions

No significant adverse reactions were noticed in vaccinated piglets on day 0 and at 1, 4, 7, and 14 days post-vaccination. 

### 3.2. Laboratory Examination

#### 3.2.1. ELISA Results

The results of the ELISA testing in the serum samples are shown in Table 2 and Figure 1. After 13 weeks, the seropositivity of all animals in the four groups was observed, possibly due to a natural infection, but the level of specific PRRSV antibodies at the end of the finishing stage (21 weeks) was significantly lower in group A in comparison to the other groups (*p* < 0.05 compared to groups B and D; *p* < 0.01 compared to group C). Significant differences between the ID vaccinated group A and the control group D were also observed at 10 and 17 weeks (*p* < 0.05). The within-group comparisons also revealed significant differences through time for all four groups (Table 2). These findings, along with the observed viral loads (Section 3.2.2), suggest that the vaccinated animals of group A had a more efficient protection, as higher antibody levels in the other groups probably indicate a more prolonged viremia and/or higher PRRSV load, which results in a more intense immune response.

#### 3.2.2. qRT-PCR Results

The results of the qRT-PCR examination of the blood samples are shown in Table 3 and Figure 2. Only 1/12 PRRSV-positive animal was found in each group at 4 weeks, and the animals of all the 4 groups were not viremic at 17 and 21 weeks. It was shown that the percentage of PRRSV-positive samples at 10 weeks was lower in group A (75%) in comparison to the other groups (91–100%). A similar observation regarding the percentage of PRRSV-positive animals in the ID vaccinated group A compared to the other groups was also observed at 7 weeks (42% for group A vs. 55–83% for the other groups) and 13 weeks (33% for group A vs. 36–58% for the other groups). Moreover, based on the qRT-PCR Ct category [1 = weak positive (Ct ≥ 35), 2 = positive (Ct < 35, ≥25), 3 = strong positive (Ct < 25)], high viremia titers were noticed in non-vaccinated groups (group C and group D) at 7, 10, and 13 weeks. The statistical analysis of qRT-PCR testing results between the groups revealed significant viral load differences only at 10 weeks in group A vs. group C (*p* < 0.01) and group B vs. group C (*p* < 0.05). The within-group comparisons also revealed significant differences through time for all four groups (Table 3). 

### 3.3. Performance Parameters

Table 4 summarizes the values for the performance parameters (the BW, age of slaughter, and ADG) per group at different periods (on admission, on weaning, at the end of the nursery period, and before their slaughter). In terms of the BW comparison at the same time between the groups, there was a statistically significantly higher BW only between the ID vaccinated group A and the non-vaccinated group and that observed before the slaughter (*p* < 0.05). The differences in the age of slaughter were not significant. The ADG was significantly higher in both vaccinated groups (A and B) in comparison to non-vaccinated groups (C and D) (*p* < 0.05).

### 3.4. Mortality

The mortality per group at different periods (nursery stage, growing stage, fattening stage) is presented in Table 5. Νο mortality was noticed in all of the groups between the vaccination time and the weaning day. The mortality was significantly lower in vaccinated groups (A and B) in comparison to non-vaccinated groups (C and D) at the finishing stage (*p* < 0.05). A similar finding regarding mortality though the whole period (*p* < 0.01) was observed. In addition, the percentage of mortality at the growing stage was significantly lower in the ID-vaccinated group A compared to that of the non-vaccinated group D (*p* < 0.05).

### 3.5. Lung Lesion Score (LLS) and Pleurisy Lesion Score (PLS)

LLS and PLS values (mean ± SD) are shown in Table 6. The results of the scoring of the lungs at the piglets’ slaughter indicated that vaccinated groups (A and B) had significantly fewer lesions and pleurisy in comparison to non-vaccinated groups (C and D) (*p* < 0.05). However, no significant differences were found between vaccinated groups A and B, as well as between the non-vaccinated groups (C and D).

### 3.6. Study Power Results

Using as an input for the G*Power software each one of the measured study variables, the study achieves more than 95% power. The protocol of the power analyses and the plots showing the study power as a function of the sample size and effect size (a quantitative measure of the magnitude of the experimental effect) are presented in Appendix A.

## 4. Discussion

The results of this study are important indications that the vaccination of piglets with the Porcilis^®^ PRRS vaccine via the intradermal route had a positive effect on their protection against PRRSV viremia, equally of a vaccine with the intramuscular application. Based on the qRT-PCR results obtained by testing the serum samples, we noticed that the percentage of viremic animals in group A was lower at 10 weeks in comparison to other groups, as well as the percentage of vaccinated groups in comparison to non-vaccinated groups at 7, 10, and 13 weeks. However, significant differences were noticed only in 10 weeks in the two vaccinated groups (A and B). The viremia titer estimation based on the Ct values revealed higher PRRSV loads in non-vaccinated groups (C and D) at 7, 10, and 13 weeks. Our results agree with previous studies, which reported that the ID vaccination with the PRRSV MLV vaccine does not affect the degree of virological protection, providing similar or even better protection compared with an IM vaccination [24,25,32]. However, these studies are reported on the vaccination of piglets at the weaning age, while the novelty of our study is the first report of ID vaccination results in younger piglets (suckling piglets at 14 days of age).

Generally, PRRSV MLV vaccines prove to be partially effective in reducing the clinical signs, viremia, and virus shedding [40]. Moreover, MLV vaccines offer consistently efficient protection against a homologous challenge, while their provided protection against a heterologous challenge is neither as consistent nor as effective [1,40,41]. The field PRRSV strain in this trial could be considered heterologous to the vaccine virus (strain DV), as the ORF 5 sequence comparison between them revealed a 90.7% sequence identity. A homology less or equal to 97–98% can be considered as a different PRRSV strain, although no defined cut-off was published [42]. Even though genetic differences between the field strain and the vaccine DV strain were evidenced, the tested PRRSV MLV vaccine provided a partial reduction in viremia in vaccinated groups (groups A and B) at 10 weeks in comparison to 7 weeks, possibly due to the infection of pigs by the field strain after the vaccine’s administration. Published controlled studies have estimated the length of viremia after a PRRSV vaccination with MLV vaccines to be 29 days on average (range 10–42 days) [25,28,43,44,45,46]. However, future studies are recommended to better understand the infection dynamics and the induced protection by vaccination.

Based on the obtained ELISA and qRT-PCR results, indications were provided that the PRRSV MLV vaccination of piglets by the ID or IM route induces an important seroconversion in 4 and 7 weeks (2–5 weeks post-vaccination) in comparison to non-vaccinated groups. In our trial, nearly 100% of pigs in vaccinated groups (groups A and B) were seropositive at week 4 (2 weeks after vaccination) while the percentage of seropositive pigs dropped significantly at week 7 and by weeks 10 and 13; all pigs in the vaccinated groups become seropositive. The possible explanation is that the vaccinated pigs were infected with a field virus, and therefore seroconverted in weeks 10 and 13. Moreover, the level of PRRSV-specific antibodies at the end of the finishing stage (21 weeks) is significantly lower in group A compared to those of other groups. However, these findings do not provide clear evidence of whether the ID-vaccinated animals of group A had a more efficient protection due to the high antibody levels in the other groups. Based on previous studies [47] and our results, the hypothesis that the transmission rate was probably lower in the ID vaccinated group A than in non-vaccinated piglets, as well as the duration of infectiousness, can be supported. Consequently, future studies are required to investigate if the ID vaccination of piglets at 14 days of age could considerably decrease the spread of PRRSV and fulfil the demands of farmers for a more efficient and cost-effective production, based on the decreased probability of piglets being infected or vaccinated piglets’ spread of the virus once they are infected [48].

The protective efficacy of a PRRS MLV vaccine is of a great economic importance due to the improvement of the health status and growth performance of vaccinated pigs [1,7,13]. Moreover, the effectiveness of a PRRS MLV vaccine must be accompanied by the absence of adverse reactions. The results of the current study proved the safety of the tested commercial ID PRRSV MLV vaccine, as no significant local and systemic reactions were noticed in vaccinated piglets until 14 days post-vaccination. Our results agree with previous studies with PRRSV MLV vaccines in gilts and weaning piglets under field conditions [24,25,28,49]. However, the literature on the safety of an ID PRRSV vaccination in suckling piglets is very limited. Previous studies with an ID vaccination in suckling piglets against other respiratory pathogens (*M. hyo*, PCV2) reported less vaccination-associated behavior changes in ID-vaccinated suckling piglets in comparison to IM-vaccinated suckling piglets, such as more active behavior and suckling activity, fewer flight reactions at the time of vaccination and less number, duration, and intensity of vocalizations [50,51]. The weakness of our study is that even if we noticed generally similar clinical observations, behavior changes in ID- and IM-vaccinated suckling piglets were not recorded and analyzed.

PRRS is characterized by a reduction in the growth performance parameters (e.g., BW and ADG) [52,53,54]. In the current study, only the ID vaccination has a significantly positive impact on the BW at the piglets’ slaughter in comparison to non-vaccinated group C, while both the ID and the IM vaccination had a significantly positive impact on the ADG in comparison to both non-vaccinated groups C and D. Previous studies have reported that a PRRSV MLV vaccination has beneficial effects on the growth performance and decreased frequency of the clinical signs in piglets exposed to PRRSV [55,56,57]. The differences in the BW and ADG observed among groups in our study were not as remarkable as previous studies had showcased [58,59,60]. This could be attributed to the short time between the vaccination and the occurrence of the natural infection, which did not allow for the improvement of the growth performance in vaccinated animals [48]. 

Mortality rates appear to be the most usually affected parameters by a PRRSV infection in piglets [5]. In our trial, the mortality rate was significantly lower in vaccinated piglets in comparison to non-vaccinated piglets at the finishing stage; the same applied to the total mortality percentage. In addition, the mortality rate was significantly lower in the vaccinated groups (A and B) than in non-vaccinated group D at the growing stage. The beneficial effects of a PRRSV vaccination on the mortality rate are correlated with our results from lung scoring. The lung scoring at the piglets’ slaughter indicated that vaccinated groups (A and B) had significantly less lung and pleurisy lesions score in comparison to non-vaccinated groups. However, no significant differences were shown between vaccinated groups A and B and between the non-vaccinated groups. Our results indicate that both the ID and IM vaccination of piglets have beneficial effects on the health status of finishing pigs, reducing the LLS and PLS parameters, mainly due to secondary bacterial co-infections.

## 5. Conclusions

Previous studies showed that an ID vaccination against PRRSV has at least a similar immunological effect compared to the IM route, providing a similar protective immunity compared with an IM vaccination (Martelli et al. 2007, 2009; Jiang et al. 2021). Moreover, these studies included trials of the vaccination of piglets at the weaning age under experimental conditions. Therefore, the novelty of our study is the first report of an ID vaccination on suckling piglets at the specific age of 14 days of age and, most importantly, under field conditions. Even though the ID vaccination of piglets against PRRSV is a common practice to the best of our knowledge, there are no published field studies investigating the effect of an ID vaccination of suckling piglets against PRRSV on the growth performance parameters up to the finishing stage.

In conclusion, this study demonstrated the safety of the ID vaccination of suckling piglets with a PRRSV MLV vaccine at 2 weeks of age under field conditions. No significant local and systemic reactions were noticed in vaccinated piglets until 14 days post-vaccination. However, our study is the first report of an ID vaccination in suckling piglets at 14 days of age. Our study indicates that both the ID and IM vaccination of piglets decreased the mortality rate, as well as the lung and pleurisy lesions score in vaccinated groups at the finishing stage. However, the ID vaccination has a positive impact on the BW at the piglets’ slaughter, while both the ID and IM vaccination had a positive impact on the ADG. 

Our results are in agreement with previous studies and provide important evidence to showcase that an ID vaccination with the PRRSV MLV vaccine does not affect the degree of virological protection, providing a similar protection compared with the IM vaccination. The aim of our field trial focused on pointing out that “vaccine protection” is not necessarily equivalent to infection avoidance. Their protective action is also associated with (i) minimizing the duration of viremia and the viral titers in the bloodstream, (ii) limiting the shedding of the virus, thus minimizing the spread to naïve animals, and (iii) minimizing the clinical impact. Even if the outcome (mortality or lung lesion scores) was not similar between IM and ID vaccinated animals, the differences in viremia are important, as they can potentially affect the shedding and infection of other animals within the farm. Additionally, as the vaccine used is attenuated, protection does not rely only on circulating antibodies, as other cell-mediated mechanisms are involved. However, such investigations or the further study of the differences following the booster immunizations of animal groups are beyond the aim of the present work.

## Figures and Tables

**Figure 1 animals-13-00061-f001:**
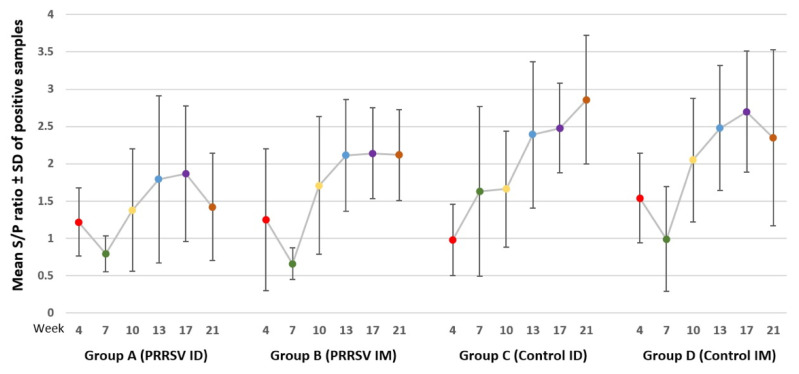
The results of ELISA examination (Mean S/P ratio ± SD) antibody-positive serum samples.

**Figure 2 animals-13-00061-f002:**
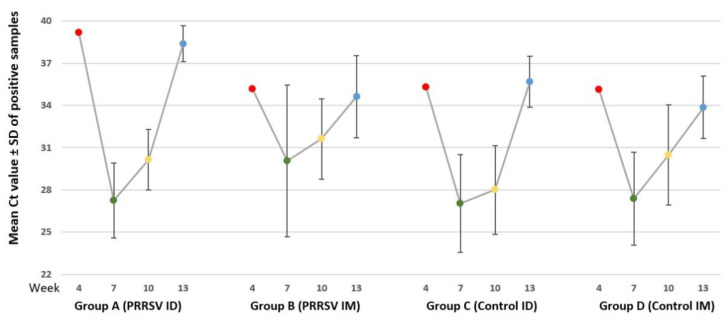
Ct values (mean ± SD) of qRT-PCR in serum samples at 4, 7, 10, and 13 weeks of age.

**Table 1 animals-13-00061-t001:** Vaccinated and non-vaccinated groups during the trial (replicates and numbers of the piglets.

Replicates	Groups			
	Group A PRRSV ID (0.2 mL Porcilis PRRS-ID)	Group B PRRSV IM (2 mL Porcilis PRRS-IM)	Group C Control ID (0.2 mL of Diluvac Forte-ID)	Group D Control IM (2 mL of Diluvac Forte-IM)
	Number of Piglets			
1	12	12	11	12
2	11	12	12	11
3	12	11	12	12
4	12	12	11	12
Total	47	47	46	47

**Table 2 animals-13-00061-t002:** ELISA testing results in serum samples at 4, 7, 10, 13, 17, and 21 weeks of age.

Group	Age (Weeks) No. of Positive/Total Mean S/P Ratio ± SD of Positive Samples					
	4	7	10	13	17	21
**Group A** (PRRSV ID)	11/12	8/12	10/12	12/12	12/12	12/12
	1.22 ± 0.46 *^b^*^,*x*^	0.79 ± 0.24 *^c^*^,*x*^	1.38 ± 0.82 *^b^*^,*y*^	1.79 ± 1.12 *^a^*^,*x*^	1.87 ± 0.91 *^a^*^,*y*^	1.42 ± 0.72 *^ab^*^,*z*^
**Group B** (PRRSV IM)	12/12	5/11 *	11/11	11/11	11/11	11/11
	1.25 ± 0.95 *^b^*^,*x*^	0.66 ± 0.21 *^c^*^,*x*^	1.71 ± 0.92 *^ab^*^,*xy*^	2.11 ± 0.75 *^a^*^,*x*^	2.14 ± 0.61 *^a^*^,*xy*^	2.12 ± 0.61 *^a^*^,*y*^
**Group C** (Control ID)	9/12	3/12	12/12	12/12	11/11 *	11/11
	0.98 ± 0.48 *^c^*^,*x*^	1.63 ± 1.14 *^c^*^,*x*^	1.66 ± 0.78 *^b^*^,*xy*^	2.39 ± 0.98 *^a^*^,*x*^	2.48 ± 0.60 *^a^*^,*xy*^	2.86 ± 0.86 *^a^*^,*x*^
**Group D** (Control IM)	8/12	7/12	12/12	12/12	12/12	12/12
	1.54 ± 0.60 *^b^*^,*x*^	0.99± 0.70 *^b^*^,*x*^	2.05 ± 0.83 *^a^*^,*x*^	2.48 ± 0.84 *^a^*^,*x*^	2.70 ± 0.81 *^a^*^,*x*^	2.35 ± 1.18 *^a^*^,*y*^

* One dead animal. *^a–c^* Comparison of ELISA testing results within groups (i.e., within a row). Values with the same superscript are not significantly different from each other (*p* ≥ 0.05), but they differ significantly (*p* < 0.05) from values with a different superscript. *^x–z^* Comparison of ELISA testing results between groups (i.e., within the column). Values with the same superscript are not significantly different from each other (*p* ≥ 0.05), but they differ significantly (*p* < 0.05) from values with a different superscript.

**Table 3 animals-13-00061-t003:** Results of qRT-PCR in serum samples at ages of 4, 7, 10, 13, 17, and 21 weeks.

Group	Age (Weeks) No. of Positive/Total Mean Ct-Value ± SD of Positive Samples No. of Positive/qRT-PCR Ct Category **				
	4	7	10	13	17
**Group A** (PRRSV ID)	1/12	5/12	9/12	4/12	0/12
	39.16 ± 0.00 *^a^*^,*x*^	27.25 ± 2.65 *^b^*^,*x*^	30.14 ± 2.14 *^b^*^,*x*^	38.40 ± 1.27 *^a^*^,*x*^	
	Cat.1: 1	Cat.2: 4, Cat.3: 1	Cat.2: 9	Cat.1: 4	
**Group B** (PRRSV IM)	1/12	6/11 *****	10/11	4/11	0/11
	35.18 ± 0.00 *^a^*^,*x*^	30.08 ± 5.39 *^b^*^,*x*^	31.62 ± 2.85 *^c^*^,*x*^	34.62 ± 2.90 *^ab^*^,*x*^	
	Cat.1: 1	Cat.1: 1, Cat. 2: 4, Cat.3: 1	Cat.1: 1, Cat.2: 9	Cat.1: 2, Cat.2: 2	
**Group C** (Control ID)	1/12	9/12	12/12	7/12	0/11 *****
	35.31 ± 0.00 *^a^*^,*x*^	27.04 ± 3.48 *^c^*^,*x*^	28.00 ± 3.15 *^c^*^,*y*^	35.69 ± 1.81 *^b^*^,*x*^	
	Cat.1: 1	Cat.2: 6, Cat.3: 3	Cat.2: 8, Cat.3: 4	Cat.1: 4, Cat.2: 3	
**Group D** (Control IM)	1/12	10/12	11/12	7/12	0/12
	35.11 ± 0.00 *^a^*^,*x*^	27.37 ± 3.31 *^c^*^,*x*^	30.49 ± 3.55 *^c^*^,*xy*^	33.87 ± 2.21 *^b^*^,*x*^	
	Cat.1: 1	Cat.2: 8, Cat.3: 2	Cat.2: 9, Cat.3: 2	Cat.1:4, Cat.2: 3	

* One dead animal; ** qRT-PCR Ct value category (Cat.), 1 = weak positive (Ct ≥35), 2 = positive (Ct < 35, ≥25), 3 = strong positive (Ct < 25). *^a–c^* Comparison of qRT-PCR testing results within groups (i.e., within row). Values with the same superscript are not significantly different from each other (*p* ≥ 0.05), but they differ significantly (*p* < 0.05) from values with a different superscript. *^x^*^,*y*^ Comparison of qRT-PCR testing results between groups (i.e., within column). Values with the same superscript are not significantly different from each other (*p* ≥ 0.05), but they differ significantly (*p* < 0.05) from values with a different superscript.

**Table 4 animals-13-00061-t004:** Performance parameters (BW, age of slaughter, ADG) (mean ± SD).

Groups	Time *	BW (kg)	Age of Slaughter (Days)	ADG (kg)
**Group A**	1	3.85 ± 0.22 *^a^*	154.50 ± 1.73 *^f^*	0.75 ± 0.01 *^g^*
	2	7.71 ± 0.38 *^b^*		
	3	27.03 ± 1.05 *^c^*		
	4	116.53 ± 1.16 *^d^*		
**Group B**	1	3.88 ± 0.19 *^a^*	156.50 ± 1.29 *^f^*	0.74 ± 0.01 *^g^*
	2	7.71 ± 0.42 *^b^*		
	3	27.05 ± 1.27 *^c^*		
	4	115.03 ± 0.43 *^de^*		
**Group C**	1	3.93 ± 0.17 *^a^*	158.75 ± 2.22 *^f^*	0.72 ± 0.01 *^h^*
	2	7.68 ± 0.44 *^b^*		
	3	26.38 ± 1.05 *^c^*		
	4	113.65 ± 0.60 *^e^*		
**Group D**	1	3.87 ± 0.03 *^a^*	160.25 ± 1.71 *^f^*	0.71 ± 0.01 *^h^*
	2	7.51 ± 0.23 *^b^*		
	3	26.23 ± 0.71 *^c^*		
	4	114.25 ± 0.96 *^de^*		

* 1: admission, 2: weaning, 3: end of nursery period, 4: slaughter. *^a–e^* Comparison of BW between groups (i.e., within the column) and at different times (i.e., 1: a, 2: b, 3: c, and 4: d–e). The same superscript indicates non-significant differences (*p* ≥ 0.05) between groups, for the same time. Different superscripts indicate significant differences (*p* < 0.05) between groups, for the same time. *^f–h^* Comparison of age of slaughter (*f*) and ADG (*g*, *h*) between groups (i.e., within the column). Values with the same superscript are not significantly different from each other (*p* ≥ 0.05), but they differ significantly (*p* < 0.05) from values with a different superscript.

**Table 5 animals-13-00061-t005:** The mortality per group at different periods.

Groups	Mortality % (No. of Dead/Total)			
	Nursery Stage	Growing Stage	Fattening Stage	Total Period
**Group A**	2.1 (1/47) *^a^*	0 (0/46) *^b^*	4.3 (2/46) *^b^*	6.3 (3/47) *^b^*
**Group B**	4.2 (2/47) *^a^*	2.29 (1/45) *^ab^*	2.3 (1/44) *^b^*	8.5 (4/47) *^b^*
**Group C**	6.5 (3/46) *^a^*	7 (3/43) *^ab^*	17.5 (7/40) *^a^*	28.3 (13/46) *^a^*
**Group D**	6.4 (3/47) *^a^*	11.4 (5/44) *^a^*	17.9 (7/39) *^a^*	32 (15/47) *^a^*

*^a^*^,*b*^ Comparison differences in mortality between groups at different periods (i.e., within column). Values with the same superscript are not significantly different from each other (*p* ≥ 0.05), but they differ significantly (*p* < 0.05) from values with a different superscript.

**Table 6 animals-13-00061-t006:** Lung lesion score (LLS) and pleurisy lesion score (PLS) at slaughter (Mean ± SD).

Groups	LLS at Slaughter	PLS at Slaughter
**Group A**	7.76 ± 2.89 *^b^*	0.06 ± 0.25 *^b^*
**Group B**	7.57 ± 2.99 *^b^*	0.06 ± 0.32 *^b^*
**Group C**	11.44 ± 6.43 *^a^*	0.46 ± 0.75 *^a^*
**Group D**	11.84 ± 7.07 *^a^*	0.45 ± 0.77 *^a^*

*^a^*^,*b*^ Comparison of LLS and PLS at slaughter between groups (i.e., within the column). Values with the same superscript are not significantly different from each other (*p* ≥ 0.05), but they differ significantly (*p* < 0.05) from values with a different superscript.

## Data Availability

Not applicable.

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
