# Peer review of "Evaluation of Intradermal PRRSV MLV Vaccination of Suckling Piglets on Health and Performance Parameters under Field Conditions"

_animals, 2022, doi:10.3390/ani13010061_

Round 1

Reviewer 1 Report

This study is very useful vaccination to prevent PRRS infection risk in pig farm. However, it will be better to revised bellow.

Introduction

Line 81: Please explain concretely how to vaccinate ID vaccination, as a needle-free administration. When vaccinate to pig as a needle-free administration, what kind of apparatus would be used etc.

Results

Line 315: Please explain clearly Study power results. I cannot understand these sentences. Are these sentences should be necessary in this study?

Author Response

Reviewer 1

We greatly appreciate your review and suggestions to improve our MS final form, please find our response

Introduction

Line 81: Please explain concretely how to vaccinate ID vaccination, as a needle-free administration. When vaccinate to pig as a needle-free administration, what kind of apparatus would be used etc.

- AU: Thank you for your comments. Please see our response on the text L81-86 (reference 26)

Results

Line 315: Please explain clearly Study power results. I cannot understand these sentences. Are these sentences should be necessary in this study?

Thank you for your comment. We have conducted G-power analysis to compute the statistical power of the analysis and sample size estimation. Therefore, we believe this sentence is a part of the statistical analysis and should not be removed from the main text.

Reviewer 2 Report

The manuscript compared the evaluation of the PRRSV MLV vaccine of suckling piglets with two different inoculation methods on health and performance parameters under field conditions. This topic is interesting and has significant instructions for the swine industry. However, the result of this study is not convincing. Some issues should be addressed and checked.

1.    The result 3.2.2 that viremia titers detected by TaqMan probe-based qRT-PCR method should be demonstrated with the virus RNA copy numbers of PRRSV, rather than the Ct values of qRT-PCR in blood samples. The test results should be converted to the number of copies using the standard curve of absolute quantitative real-time PCR.

2.    The use of adverbs should be careful, for instance, the “very” in line 135.

3.    Check the format of Table 1.

4.    The results from different groups should be exhibited on one figure with different colors. For readers, the differences between groups are more intuitive. (Figures 1 and 2)

5.    Why did the authors consider that the level of specific PRRSV antibodies was significantly lower in group A in comparison to the other groups indicating group A had more efficient protection? (3.2.1)

Author Response

Reviewer 2

We greatly appreciate your review and suggestions to improve our MS final form, please find our response

The manuscript compared the evaluation of the PRRSV MLV vaccine of suckling piglets with two different inoculation methods on health and performance parameters under field conditions. This topic is interesting and has significant instructions for the swine industry. However, the result of this study is not convincing. Some issues should be addressed and checked

  1. The result 3.2.2 that viremia titers detected by TaqMan probe-based qRT-PCR method should be demonstrated with the virus RNA copy numbers of PRRSV, rather than the Ct values of qRT-PCR in blood samples. The test results should be converted to the number of copies using the standard curve of absolute quantitative real-time PCR.

- AU: Thank you for your comments. Undoubtedly, providing information regarding the viral RNA genomic copies/μl in each tested specimen would constitute the most accurate approach to PRRSV quantification. However, we do not possess such quantified standards to generate the requested standard curve, and thus, the assay is not adapted to be used for the absolute quantification of PRRSV. Therefore, as an alternative, the obtained Ct values were used as viral load estimates. This comprises a practice widely accepted and used for viral load monitoring in clinical specimens of animals and humans.

  1. The use of adverbs should be careful, for instance, the “very” in line 135.

- AU: Thank you for your comments. Please check our response on the text L135.

  1. Check the format of Table 1.

- AU: Thank you for your comments. Please check our response on the text

  1. The results from different groups should be exhibited on one figure with different colors. For readers, the differences between groups are more intuitive. (Figures 1 and 2)

- AU: Thank you for your comments. Please check our response on the text

  1. Why did the authors consider that the level of specific PRRSV antibodies was significantly lower in group A in comparison to the other groups indicating group A had more efficient protection? (3.2.1)

- AU: Thank you for your comment. Please note, that the “more efficient protection” suggested for group A is based on the comparative interpretation not only of antibody levels but also of viremia loads. It is evident that the viremia titers in Group A were lower at W13, which indicates a considerably more effective viral clearance/minimization of viremia, compared to what was observed in other groups. In addition, as the animals of the farm were naturally infected, it is expected that prolonged exposure of previously immunized animals to the wild-type viral strain will induce anamnestic immune responses that generate higher antibody levels

Reviewer 3 Report

Summary of paper:

The authors provided real time data from a field  studying using the PRRSV vaccination given to piglets at 2 weeks of age using intradermal route of administration vs. intramuscular route of administration. The hypothesis is that an ID vaccine would provide better protection and the reporting outcomes were: 1) seropositivity, piglets characteristics at age of slaughter, mortality rate and LLS and PLS  scores at time of slaughter. The only significant finding of this paper was a difference in viral titer for the group that got the vaccine using ID route (group A), however that did not impact the outcomes significantly when compared with the IM route of administration. Therefore, this paper provides a summary of the findings that the ID route is as effective as suggested by previous findings. The role of lower viremia remains unclear since the outcomes did not change. The reviewer suggests the following to the authors:

1- Is this sentence chopped? Line 64-65

PRRS control by vaccination is important also to decrease eco-64 nomic losses due to decreased performance and feed efficiency due.

2. In the material and methods section

The materials and methods section states there were two unique number and unique colors for ID’ing the various piglets. The research included four groups. The methods state that control vs. experimental was labeled by different color, how did the researchers differentiate between the IM vs. ID groups?

3- In figure 2, It is clear that it took more cycles at week 4 and week 13  for group A to see viral titers in the random sample, but then when you look at  table 2, it does not generate a lot of meaning when comparing group A and B. From the CT values, it suggests that group A has less viral tiers, but when looking at the S/P ratio, it did not impact it. Basically, viremia did not impact S/P ratio, so what is the significance of looking at viremia when it did not create a difference in viral protection based on route of administration? Can this be addressed in the discussion.

4- The overall outcome on piglets were equal based on the data provided. The only difference suggested by the results is lower viral titers, however, this did not change the outcome to better protection as stated several times in the paper. Also, the vaccine seems to be protective for 3 weeks only. Should the study consider repeating vaccination and observing a change in outcomes that demonstrate a difference between ID and IM routes. For now, the S/P ratio is not different between IM and ID routes, the protection from death is not different either, the LLS or PLS at time of laughter are not different between IM and ID. The only difference is viremia, but it does not impact the outcome. The conclusion of the paper should be an alternative route of administration is as effective as IM route and perhaps address why IM route is not the best route for some piglets in the discussion.  

5- Line 324, remove better, it is equal protection.

6- The papers conclusion Line 411-414 is misleading. Not only ID provided the protection IM as well, I suggest revising the conclusion to showcase the results.

Author Response

Reviewer 3

We greatly appreciate your review and suggestions to improve our MS final form, please find our response

Summary of paper:

The authors provided real time data from a field  studying using the PRRSV vaccination given to piglets at 2 weeks of age using intradermal route of administration vs. intramuscular route of administration. The hypothesis is that an ID vaccine would provide better protection and the reporting outcomes were: 1) seropositivity, piglets characteristics at age of slaughter, mortality rate and LLS and PLS  scores at time of slaughter. The only significant finding of this paper was a difference in viral titer for the group that got the vaccine using ID route (group A), however that did not impact the outcomes significantly when compared with the IM route of administration. Therefore, this paper provides a summary of the findings that the ID route is as effective as suggested by previous findings. The role of lower viremia remains unclear since the outcomes did not change. The reviewer suggests the following to the authors:

1- Is this sentence chopped? Line 64-65

PRRS control by vaccination is important also to decrease eco-64 nomic losses due to decreased performance and feed efficiency due.

- AU: Please check our response to the text L65.

  1. In the material and methods section

The materials and methods section states there were two unique number and unique colors for ID’ing the various piglets. The research included four groups. The methods state that control vs. experimental was labeled by different color, how did the researchers differentiate between the IM vs. ID groups?

- AU: Please check our response to the text L148.

3- In figure 2, It is clear that it took more cycles at week 4 and week 13  for group A to see viral titers in the random sample, but then when you look at  table 2, it does not generate a lot of meaning when comparing group A and B. From the CT values, it suggests that group A has less viral tiers, but when looking at the S/P ratio, it did not impact it. Basically, viremia did not impact S/P ratio, so what is the significance of looking at viremia when it did not create a difference in viral protection based on route of administration? Can this be addressed in the discussion.

4- The overall outcome on piglets were equal based on the data provided. The only difference suggested by the results is lower viral titers, however, this did not change the outcome to better protection as stated several times in the paper. Also, the vaccine seems to be protective for 3 weeks only. Should the study consider repeating vaccination and observing a change in outcomes that demonstrate a difference between ID and IM routes. For now, the S/P ratio is not different between IM and ID routes, the protection from death is not different either, the LLS or PLS at time of laughter are not different between IM and ID. The only difference is viremia, but it does not impact the outcome. The conclusion of the paper should be an alternative route of administration is as effective as IM route and perhaps address why IM route is not the best route for some piglets in the discussion. 

- AU: Thank you for your comments. We want to point out that “vaccine protection” is not necessarily equivalent to infection avoidance. Their protective action is also associated with i) minimizing the duration of viremia and the viral titers in the bloodstream, ii) limiting shedding of the virus, thus minimizing spread to naïve animals, and iii) minimizing clinical impact. Even if not the outcome (death or lesion scores) was similar between IM and ID vaccinated animals, the differences in viremia are important, as they can potentially affect the shedding and infection of other animals within the farm. Additionally, as the vaccine used is attenuated, protection does not rely only on circulating antibodies, as other cell-mediated mechanisms are involved. However, such investigations or the further study of differences following booster immunizations of animal groups are beyond the aim of the present work.

Please check our response on the text L324.

5- Line 324, remove better, it is equal protection.

- AU: Thank you for your comments. Please check our response on the text L324.

6- The papers conclusion Line 411-414 is misleading. Not only ID provided the protection IM as well, I suggest revising the conclusion to showcase the results.

- AU: Thank you for your comments. Please check our response on the text L408-440.

Reviewer 4 Report

This paper describes a trial that compares the intradermal (ID) administration of a commercial PRRSV-1 modified live virus (MLV) vaccine to the intramuscular (IM) administration of the same vaccine in relation to their effect on health and performance of the vaccinees. An overall sample of 187 suckling piglets of a PRRSV-positive commercial farrow-to-finish farm was divided into 4 groups of 46 or 47 animals each: 1) group PRRSV vaccine ID, 2) group of PRRSV vaccine IM, 3) group of control adjuvant alone ID and 4) control adjuvant alone IM. All these 4 treatments were applied in these animals at 2 weeks of age. Collected blood serum samples were tested by ELISA and qRT-PCR. Side effects, body weight (BW), average daily gain (ADG), mortality rate, and lung and pleurisy lesions scores (LLS, PLS) were also recorded. Serologic conversion is detected in all animals at 4 and 7 weeks. Significant differences in qRT-PCR results were noticed only at 10 weeks in group 1 and 2 vs control groups. The authors conclude that ID vaccination of suckling piglets with a PRRSV-1 MLV vaccine has protective effect similar to that obtained by IM injection thus achieving a positive effect on piglet health and performance

Since the initial experiments supported by the vaccine company ( Intervet > > MSD)  that developed the Porcilis strain of MLV anti PRRSV-1 vaccine ( reported initially by Dr Martelli et al in 2006/2007 ) it has been known that the ID route is an effective strategy of PRRSV MLV vaccination that minimizes/eliminates the damage caused by the needle at the muscular level, prevents carry over iatrogenic ( animal to animal via shared needle) infections and permits optimal immunization at the level of skin that is  rich in antigen-presenting cells.  

Undoubtedly the concept of ID vaccination X PRRSV has been a solid contribution to vaccinology.

 Throughout multiple publications in the last two decades it became demonstrated, time and time again, that the Porcilis ID route of vaccination is a practical and effective strategy of immunization. Those multiple experiments have been (just like the present manuscript) financed and supported in different ways by the MSD company, which is the exclusive developer/marketer  of this Porcilis strain. In all previous reports it was concluded that the ID route matched, if not enhanced, the efficacy of the IM administration, which is an additional plus of versatility for the application of Porcilis under different scenarios. This current manuscript reiterates such concept. In fact such comparative conclusion (i.e that the ID route being as safe as the IM route) is the only valid (although not novel) take-home message that I can find in this paper. The rest of the discussion centers on attempting to demonstrate through multiple parameters or correlates that the ID/IM vaccination of suckling pigs can be equally effective in “curing” the endemic PRRSV infection that affects the breeding stock and spreads to all segments of production of this farm. Unfortunately the design selected by the authors ( vaccine “field study design” followed up by serology and PCR evaluation of viremia) does not permit to reach any valid conclusion regarding true effectiveness of the therapeutic application of MLV vaccine due to the inherent draws of the design itself: i.e. inability to ascertain what comes first into the body  (if if wt PRSSV or MLV), inability to ascertain if seroconversion ( or lack thereof) is due to wt PRRSV infection or vaccine response, etc). A different more conclusive approach is when the vaccine study design is set as a true preventive or prophylactic approach of the vaccine ( prior vaccination of PRRSv negative piglets followed by experimental controlled challenge with wt PRRSV ( as reported by reference 32 cited in this paper , which is another of the many MSD supported reports that have been already published).  

Author Response

- AU: Thank you for your comments.

Previous studies showed that ID vaccination against PRRSV has at least a similar immunological effect compared to IM, providing similar protective immunity compared with IM vaccination (Martelli et al. 2007, 2009; Jiang et al. 2021). Moreover, these studies included trials of vaccination of piglets at weaning age under experimental conditions. Therefore, the novelty of our study is the first report of ID vaccination on suckling piglets at the specific age of 14 days of age and most importantly under field conditions. Even though ID vaccination of piglets against PRRSV is a common practice to the best of our knowledge, there are no published field studies investigating the effect of ID vaccination of suckling piglets against PRRSV on growth performance parameters up to the finishing stage.

Please refer to our response on the text L408-417.

Round 2

Reviewer 3 Report

N/A

Reviewer 4 Report

Despite the authors' revisions and thoughtful responses to this reviewer's comments, my opinion on this piece of research as is described in this revision continues to be negative. I consider that "field evaluations designs" like the one described in this manuscript, although may be commonly used in comercial pamphlets or blog testimonials to support products produced by biological companies, are ineffective to prove the true comparative immunogenic efficacy of vaccines more rigorously. For details on the rationale of this reviewer's opinion please see comments prided in the review of the original manuscript (report1).